# PTaRL: Prototype-based Tabular Representation Learning via Space Calibration

**Hangting Ye**[1], **Wei Fan**[2], **Xiaozhuang Song**[3], **Shun Zheng**[4], **He Zhao**[5], **Dandan Guo**[1]*, **Yi Chang**[1]*
School of Artificial Intelligence, Jilin University[1] University of Oxford[2]
The Chinese University of Hong Kong, Shenzhen[3] Microsoft Research[4] CSIRO's Data61[5]
`yeht2118@mails.jlu.edu.cn`, `wei.fan@wrh.ox.ac.uk`,
`xiaozhuangsong1@link.cuhk.edu.cn`, `shun.zheng@microsoft.com`,
`he.zhao@ieee.org`, `{guodandan,yichang}@jlu.edu.cn`

## Abstract

Tabular data have been playing a mostly important role in diverse real-world fields, such as healthcare, engineering, finance, etc. With the recent success of deep learning, many tabular machine learning (ML) methods based on deep networks (e.g., Transformer, ResNet) have achieved competitive performance on tabular benchmarks. However, existing deep tabular ML methods suffer from the *representation entanglement* and *localization*, which largely hinders their prediction performance and leads to *performance inconsistency* on tabular tasks. To overcome these problems, we explore a novel direction of *applying prototype learning for tabular ML* and propose a prototype-based tabular representation learning framework, PTaRL, for tabular prediction tasks. The core idea of PTaRL is to construct prototype-based projection space (P-Space) and learn the disentangled representation around global data prototypes. Specifically, PTaRL mainly involves two stages: (i) Prototype Generation, that constructs global prototypes as the basis vectors of P-Space for representation, and (ii) Prototype Projection, that projects the data samples into P-Space and keeps the core global data information via Optimal Transport. Then, to further acquire the disentangled representations, we constrain PTaRL with two strategies: (i) to diversify the coordinates towards global prototypes of different representations within P-Space, we bring up a diversification constraint for representation calibration; (ii) to avoid prototype entanglement in P-Space, we introduce a matrix orthogonalization constraint to ensure the independence of global prototypes. Finally, we conduct extensive experiments in PTaRL coupled with state-of-the-art deep tabular ML models on various tabular benchmarks and the results have shown our consistent superiority.

## 1 Introduction

Tabular data, usually represented as tables in a relational database with rows standing for the data samples and columns standing for the feature variables (e.g., categorical and numerical features), has been playing a more and more vital role across diverse real-world fields, including healthcare (Hernandez et al., 2022), engineering (Ye et al., 2023), advertising (Frosch et al., 2010), finance (Assefa et al., 2020), etc. Starting from traditional machine learning methods (e.g., linear regression (Su et al., 2012), logistic regression (Wright, 1995)) to tree-based methods (e.g. XGBoost (Chen & Guestrin, 2016), LightGBM (Ke et al., 2017)), tabular machine learning has received broad attention from researchers for many decades.

More recently, with the great success of deep networks in computer vision (CV) (He et al., 2016) and natural language processing (NLP) (Devlin et al., 2018), numerous methods based on deep learning have been proposed for tabular machine learning (ML) to accomplish tabular prediction tasks (Song et al., 2019; Huang et al., 2020; Gorishniy et al., 2021; Wang et al., 2021). For example, (Song et al., 2019) proposed AutoInt based on transformers, (Gorishniy et al., 2021) further improved

---

*Corresponding authors.

AutoInt through better token embeddings, and (Wang et al., 2021) proposed DCN V2 that consists of an MLP-like module and a feature crossing module.

While the recent deep learning solutions have performed competitively on tabular benchmarks (Gorishniy et al., 2021; Shwartz-Ziv & Armon, 2022), there still exists an *performance inconsistency* in their predictions on tabular tasks: existing state-of-the-art deep tabular ML models (e.g., FT-Transformer (Gorishniy et al., 2021), ResNet (Gorishniy et al., 2021)) cannot perform consistently well on different tasks, such as regression, classification, etc. We investigate the learned patterns of deep tabular ML models and identify two inherent characteristic hindering prediction: (i) *representation entanglement*: the learned representations of existing deep tabular methods are usually entangled and thus cannot support clear and accurate decision-making, and (ii) *representation localization*: each data sample are represented distinctively, making the global data structures over data samples are overlooked.

To better overcome the aforementioned challenges, we explore the direction of *applying prototype learning for tabular modeling*, and accordingly we propose PTARL, a prototype-based tabular representation learning framework for tabular ML predictions. The core idea of PTARL is to develop a *prototype-based projection space* (P-Space) for deep tabular ML models, in which the disentangled representations[1] around pre-defined global prototypes can be acquired with global tabular data structure to enhance the tabular predictions. Specifically, our PTARL mainly involves two stages, (i) Prototype Generating and (ii) Representation Projecting. In the first stage, we construct $K$ global prototypes for tabular representations, each of which is regarded as the basis vector for the P-Space to stimulate disentangled learning for more global data representations. We initialize the global prototypes with *K-Means clustering* (Hartigan & Wong, 1979) to facilitate the efficiency of prototype learning. In the second stage, we project the original data samples into P-Space with the global prototypes to learn the representations with global data structure information. To learn the global data structure, we propose a shared estimator to output the projected representations with global prototypes; besides, we propose to utilize *Optimal Transport* (Peyré et al., 2017) to jointly optimize the learned representations in P-Space with global prototypes and original representations by deep tabular models, to preserve original data structure information.

In addition to employing global prototypes, we propose two additional strategies to further disentangle the learned representations in PTARL: (i) Coordinates Diversifying Constraint motivated by contrastive learning that diversifies the representation coordinates of data samples in P-Space to represent data samples in a disentangled manner, and (ii) Matrix Orthogonalization Constraint that makes the defined global prototypes in P-Space orthogonal with each other to ensure the independence of prototypes and facilitate the disentangled learning. In brief, our contribution can be summarized as follows:

- We investigated the learned patterns of deep tabular models and explore a novel direction of *applying prototype learning for tabular machine learning* to address representation entanglement and localization.

- We propose a model-agnostic prototype-based tabular representation learning framework, PTARL for tabular prediction tasks, which transforms data into the prototype-based projection space and optimize representations via Optimal Transport.

- We propose two different strategies, the Coordinates Diversifying Constraint and the Matrix Orthogonalization Constraint to make PTARL learn disentangled representations.

- We conducted extensive experiments in PTARL coupled with state-of-the-art (SOTA) deep tabular ML models on various tabular benchmarks and the comprehensive results along with analysis and visualizations demonstrate our effectiveness.

## 2 RELATED WORK

**Deep Learning for Tabular machine learning.** Starting from statistical machine learning methods (e.g., linear regression (Su et al., 2012), logistic regression (Wright, 1995)) to tree-based methods

---

[1]In our paper, "disentangled representations" means the representations are more separated and discriminative for supervised tabular modeling tasks, which is different from disentanglement in deep generative models.

(e.g. XGBoost (Chen & Guestrin, 2016), LightGBM (Ke et al., 2017)), traditional machine learning methods are broadly used for tabular machine learning. More recently, inspired by the success of of deep learning in computer vision (CV) (He et al., 2016) and natural language processing (NLP) (Devlin et al., 2018), numerous methods based on deep learning have been proposed for tabular machine learning to accomplish tabular prediction tasks (Song et al., 2019; Huang et al., 2020; Gorishniy et al., 2021; Wang et al., 2021). Among these works, Wang et al. (2021) proposed DCN V2 that consists of an MLP-like module and a feature crossing module; AutoInt (Song et al., 2019) leveraged the Transformer architecture to capture inter-column correlations; FT-Transformer (Gorishniy et al., 2021) further enhanced AutoInt's performance through improved token embeddings; ResNet for tabular domain (Gorishniy et al., 2021) also achieved remarkable performance. However, these methods may fail to capture the global data structure information, and are possibly affected by the representation coupling problem. Therefore, they cannot perform consistently well on different tasks, e.g. regression and classification. Recently, another line of research has tried to use additional information outside target dataset to enhancing deep learning for tabular data. TransTab (Wang & Sun, 2022) incorporates feature name information into Transformer to achieve cross table learning. XTab (Zhu et al., 2023) pretrains Transformer on a variety of datasets to enhance tabular deep learning. Different from this line, PTARL does not need additional information outside target dataset. Note that PTARL, as a general representation learning pipeline, is model-agnostic such that it can be integrated with many of the above deep tabular ML models to learn better tabular data representations.

**Prototype Learning.** Typically, a prototype refers to a proxy of a class and it is computed as the weighted results of latent features of all instances belonged to the corresponding class. Prototype-based methods have been widely applied in a range of machine learning applicaitons, like computer vision (Yang et al., 2018; Li et al., 2021; Nauta et al., 2021; Zhou et al., 2022), natural language processing (Huang et al., 2012; Devlin et al., 2018; Zalmout & Li, 2022). In the field of CV, prototype learning assigns labels to testing images by computing their distances to prototypes of each class. This method has been proven to make model to be more resilient and consistent when dealing with few-shot and zero-shot scenarios (Yang et al., 2018). Likewise, in the field of natural language processing (NLP), taking the mean of word embeddings as prototypes for sentence representations has also demonstrated robust and competitive performance on various NLP tasks.

The mentioned approachs generally employ the design of prototype learning to facilitate the sharing of global information across tasks, enabling rapid adaptation of new tasks (Huang et al., 2012; Hoang et al., 2020; Li et al., 2021; Zhou et al., 2022). Similarly, in tabular deep learning, the global information of data samples is crucial for inferring labels of each data sample (Zhou et al., 2020; Du et al., 2021). This inspired us to incorporate prototype learning into our proposed framework for capturing global information and leveraging it to enhance the tabular learning performance.

## 3 BACKGROUND

**Notation.** Denote the $i$-th sample as $(x_i, y_i)$, where $x_i = (x_i^{(num)}, x_i^{(cat)}) \in \mathbb{X}$ represents numerical and categorical features respectively and $y_i \in \mathbb{Y}$ is the corresponding label. A tabular dataset $D = \{X, Y\}$ is a collection of $n$ data samples, where $X = \{x_i\}_{i=1}^n$ and $Y = \{y_i\}_{i=1}^n$. We use $D_{train}$ to denote training set for training, $D_{val}$ to denote validation set for early stopping and hyperparameter tuning, and $D_{test}$ to denote test set for final evaluation. Note that in this paper we consider deep learning for supervised tabular prediction tasks: binary classification $\mathbb{Y} = \{0, 1\}$, multiclass classification $\mathbb{Y} = \{1, ..., c\}$ and regression $\mathbb{Y} = \mathbb{R}$. The goal is to obtain an accurate deep tabular model $F(\cdot; \theta) : \mathbb{X} \to \mathbb{Y}$ trained on $D_{train}$.

**Optimal Transport.** Although Optimal Transport (OT) possesses a rich theoretical foundation, we focus our discussion solely on OT for discrete probability distributions, please refer to (Peyré et al., 2017) for more details. Let us consider $p$ and $q$ as two discrete probability distributions over an arbitrary space $\mathbb{S} \in \mathbb{R}^d$, which can be expressed as $p = \sum_{i=1}^n a_i \delta_{x_i}$ and $q = \sum_{j=1}^m b_j \delta_{y_j}$. In this case, $\boldsymbol{a} \in \sum^n$ and $\boldsymbol{b} \in \sum^m$, where $\sum^n$ represents the probability simplex in $\mathbb{R}^n$. The OT distance between $p$ and $q$ is defined as:

$$\text{OT}(p, q) = \min_{\mathbf{T} \in \Pi(p,q)} \langle \mathbf{T}, \mathbf{C} \rangle, \tag{1}$$

where $\langle \cdot, \cdot \rangle$ is the Frobenius dot-product and $\mathbf{C} \in \mathbb{R}_{\geq 0}^{n \times m}$ is the transport cost matrix constructed by $C_{ij} = Distance(x_i, y_j)$. The transport probability matrix $\mathbf{T} \in \mathbb{R}_{\geq 0}^{n \times m}$, which satisfies $\Pi(p, q) := \{\mathbf{T} | \sum_{i=1}^{n} T_{ij} = b_j, \sum_{j=1}^{m} T_{ij} = a_i\}$, is learned by minimizing $\langle \mathbf{T}, \mathbf{C} \rangle$. Directly optimizing Eq. 1 often comes at the cost of heavy computational demands, and OT with entropic regularization is introduced to allow the optimization at small computational cost in sufficient smoothness (Cuturi, 2013).

# 4 PROPOSED METHOD: PTARL

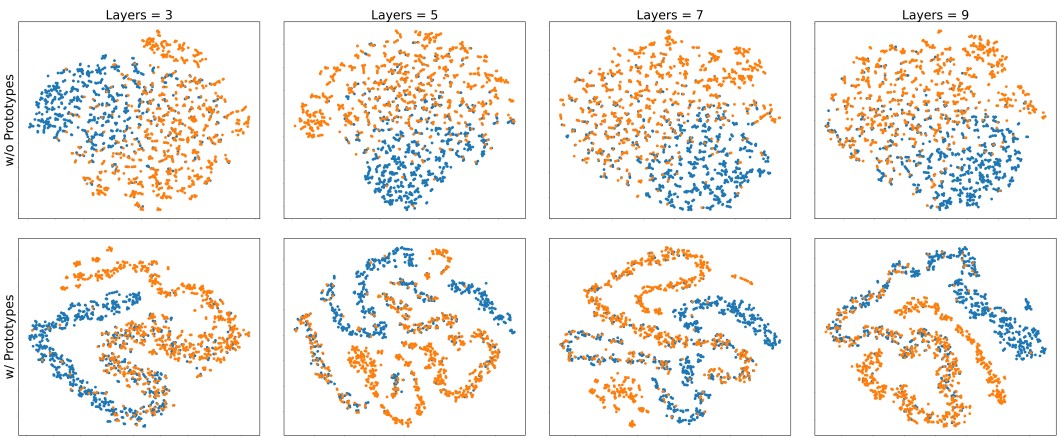

Figure 1: The visualization of representations of deep network w/o and w/ PTARL with varying model layer depths.

## 4.1 MOTIVATION AND OVERALL PIPELINE

In the context of tabular data, the intrinsic heterogeneity presents a challenge for achieving satisfactory performance using deep models. As shown in the first subfigure of Fig. 1, the latent representations learned by the FT-Transformer (Gorishniy et al., 2021) (one of the SOTA deep tabular models) on a binary classification dataset Adult (Kohavi et al., 1996) are entangled. To validate whether it is caused by the limitation of model capacity, we gradually increase FT-Transformer's layer depths and visualize the corresponding latent representation by T-SNE (Van der Maaten & Hinton, 2008). As shown in the first row of Fig. 1, with a sequential increase in the number of model layers, we could observe the *representation entanglement* phenomenon has not be mitigated while gradually augmenting the model capacity. In addition, our empirical observations indicate that this phenomenon also exists for other deep tabular models and we recognize that the representation entanglement is the inherent limitation of deep models in tabular domain. Moreover, the learned representations also lack global data structure information, which failed to model the shared information among the data instances. Compared to other domains like CV and NLP, especially in the heterogeneous tabular domain, samples overlook the statistical global structure information among the total dataset would drop into *representation localization*. Furthermore, recent researches (Gorishniy et al., 2021; Shwartz-Ziv & Armon, 2022) show that different types of data may require varying types of deep models (e.g. Transformer based and MLP based architecture).

To address the aforementioned limitations of deep models for the tabular domain, we apply prototype learning into tabular modeling and propose the prototype-based tabular representation learning (PTARL) framework. Note that PTARL, as a general representation learning framework, is model-agnostic such that it can be coupled with any deep tabular model $F(\cdot; \theta)$ to learn better tabular data representations in our redefined *Prototype-based Projection Space*, which is the core of PTARL. In the following, we will elaborate on the specific learning procedure of our PTARL in Section 4.2; then, we will provide two constraints to further constraint PTARL for representation calibration and prototype independence in Section 4.3. As shown in the second row of Fig. 1, with PTARL, the latent space is calibrated to make the representation disentangled. Fig. 2 gives an overview of the proposed PTARL. Before the introduction of PTARL, we first present the formal definition of the prototype-based projection space as follows:

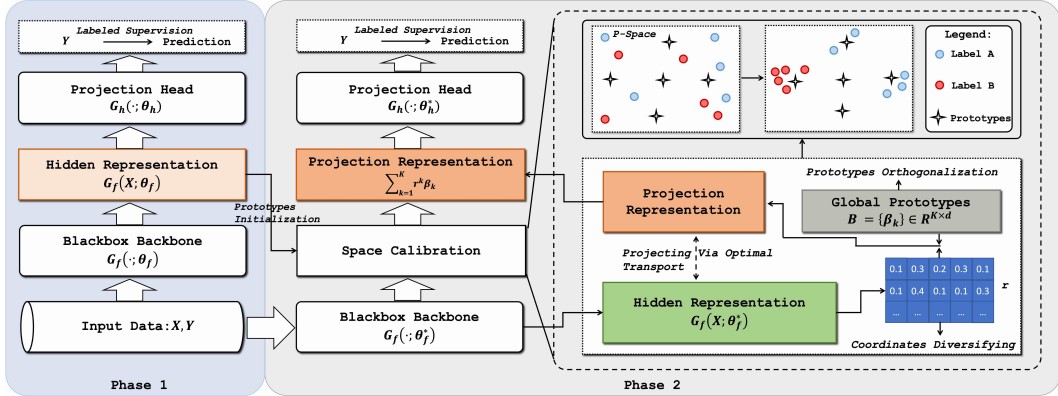

Figure 2: The PTARL framework. The original representation of each sample by backbone would be pushed forward to the corresponding projection representation by minimizing the Optimal Transport Distance. The two sentences "coordinates diversifying" and "prototypes orthogonalization" correspond to two constraints for representation disentanglement.

**Definition 1. Prototype-based Projection Space (P-Space).** Given a set of global prototypes $\mathcal{B} = \{\beta_k\}_{k=1}^{K} \in \mathbb{R}^{K \times d}$ where $K$ is the number of prototypes and $\beta_k$ is the representation of the $k$-th prototype with $d$ as the hidden dimension, we define the P-Space as a projection space consisting of the global prototypes $\mathcal{B}$ as the basis vectors for representation. For example, given a representation in P-Space with **_coordinates_** denoted as $r = \{r^k\}_{k=1}^{K} \in \mathbb{R}^{K \times 1}$, each representation in P-Space can be formulated as $\sum_{k=1}^{K} r^k \beta_k$.

## 4.2 PROTOTYPE-BASED TABULAR REPRESENTATION LEARNING (PTARL)

PTARL consists of the following two stages: (i) prototype generating, that constructs global prototypes of P-Space, and (ii) representation projecting, that projects $F(\cdot; \theta)$'s representations into P-Space to capture global data structure information via optimal transport.

**Global Prototypes Initialization.** To start with, let us rewrite the deep tabular network $F(\cdot; \theta)$ as $G_h(G_f(\cdot; \theta_f); \theta_h)$, where $G_f(\cdot; \theta_f)$ is the backbone and $G_h(\cdot; \theta_h)$ is the head parameterized by $\theta_f$ and $\theta_h$ respectively. To obtain global prototypes initialization, we first train $F(\cdot; \theta)$ by:

$$\min \mathcal{L}_{task}(X, Y) = \min_{\theta_f, \theta_h} \mathcal{L}(G_h(G_f(X; \theta_f); \theta_h), Y), \tag{2}$$

where $\mathcal{L}$ is the loss function, and then propose applying K-Means clustering (Hartigan & Wong, 1979) to the output of the trained backbone $G_f(X; \theta_f)$:

$$\min_{C \in \mathbb{R}^{K \times d}} \frac{1}{n} \sum_{i=1}^{n} \min_{\tilde{y}_i \in \{0,1\}^K} \|G_f(x_i; \theta_f) - \tilde{y}_i^{\mathrm{T}} C\|, \text{ subject to } \tilde{y}_i^{\mathrm{T}} \mathbf{1}_K = 1, \tag{3}$$

where $\mathbf{1}_K \in \mathbb{R}^K$ is a vector of ones, $\tilde{y}_i$ is the centroid index and $C$ is the centroid matrix. The centroid matrix $C$ would be served as the initialization of prototypes $\mathcal{B}$ to guide the stage (ii) training process of PTARL. The advantage of using K-Means to generate the initialization of global prototypes lies in (a) enabling the preservation of global data structure information from the trained $F(\cdot; \theta)$ in stage (i), despite the presence of noise, and (b) facilitating a faster convergence of the stage (ii) training process compared to random initialization.

**Representation Projecting with Global Data Information via Optimal Transport.** After constructing the global prototypes $\mathcal{B}$, we re-initialize the parameters of $F(\cdot; \theta)$ and start to project $F(\cdot; \theta)$'s representation into P-Space via PTARL to capture global data structure information. To obtain the projection of $i$-th instance representation $G_f(x_i; \theta_f)$, we use a shared estimator $\phi(\cdot; \gamma)$ with learnable $\gamma$ to calculate its coordinate $r_i$ by $\phi(G_f(x_i; \theta_f); \gamma)$. Mathematically, our formulation for the $i$-th projection representation distribution takes the following form: $Q_i = \sum_{k=1}^{K} r_i^k \delta_{\beta_k}$. On the other hand, the $i$-th latent representation $G_f(x_i; \theta_f)$ could be viewed as the empirical distribution over single sample representation: $P_i = \delta_{G_f(x_i; \theta_f)}$.

Since all samples are sampled from the same data distribution, it is reasonable to assume that there exists shared structural information among these samples. To capture the shared global data structure

information, we formulate the representation projecting as the process of extracting instance-wise data information by $G_f(x_i; \theta_f)$ to $P_i$, and then pushing $P_i$ towards $Q_i$ to encourage each prototype $\beta_k$ to capture the shared *global data structure information*, a process facilitated by leveraging the optimal transport (OT) distance:

$$\min \mathcal{L}_{projecting}(X, \mathcal{B}) = \min \frac{1}{n} \sum_{i=1}^{n} \text{OT}(P_i, Q_i) = \min_{\theta_f, \gamma, \mathcal{B}} \frac{1}{n} \sum_{i=1}^{n} \min_{\mathbf{T}_i \in \Pi(P_i, Q_i)} \langle \mathbf{T}_i, \mathbf{C}_i \rangle, \quad (4)$$

where $\mathbf{C}$ is the transport cost matrix calculated by cosine distance and $\mathbf{T}$ is the transport probability matrix, please refer to Section 3 for more details about OT. We provide the more detailed explanation of the optimization process of Eq. 4 in Appendix A.8. In contrast to the latent space of $F(\cdot; \theta)$, the P-Space consists of explicitly defined basis vectors denoted as global prototypes, thus it is more accurate to give predictions on representations within P-Space. We incorporate the label information by re-defining Eq. 2:

$$\min \mathcal{L}_{task}(X, Y) = \min_{\theta_f, \theta_h, \gamma, \mathcal{B}} \frac{1}{n} \sum_{i=1}^{n} \mathcal{L}(G_h(\sum_{k=1}^{K} r_i^k \beta_k; \theta_h), y_i), \quad (5)$$

By representing each projected representation using the shared basis vectors (global prototypes $\mathcal{B}$) within the P-Space, we can effectively model this *global data structure information* to effectively solve one of the deep tabular network's inherent characteristic: sample localization. The representation entanglement problem still exists within P-Space, and we will introduce the designed constraints to make the representation disentangled in the following section.

### 4.3 CONSTRAINTS FOR PTARL VIA COORDINATES DIVERSIFYING AND PROTOTYPE MATRIX ORTHOGONALIZATION

**P-Space Coordinates Diversifying Constraint**. Directly optimizing the above process could potentially result in a collapse of coordinates generating, which entails all projection representations exhibiting similar coordinates within P-Space. To alleviate it, we design a constraint to diversifying the coordinates towards global prototypes within P-Space to separate the representation into several disjoint regions, where representations with similar labels are in the same regions. Specifically, for classification, we categorize samples within the same class as positive pairs (Khosla et al., 2020), while those from different classes form negative pairs. In the context of regression, we segment the labels within a minibatch into $t$ sub-intervals as indicated by $t = 1 + \log(n_b)$, where $n_b$ is bachsize. Samples residing within the same bin are regarded as positive pairs, while those from distinct bins are regarded as negative pairs, thereby facilitating the formation of pairings. This process is achieved by minimizing:

$$\mathcal{L}_{diversifying}(X) = -\sum_{i=1}^{n_b} \sum_{j=1}^{n_b} \mathbf{1}\{y_i, y_j \in \text{positive pair}\} \log \frac{\exp(\cos(r_i, r_j))}{\sum_{i=1}^{n_b} \sum_{j=1}^{n_b} \exp(\cos(r_i, r_j))} \quad (6)$$

This constraint is motivated by contrastive learning (CL) (Khosla et al., 2020). Distinct from conventional contrastive learning methods in tabular domain that directly augment the variety of sample representations (Wang & Sun, 2022; Bahri et al., 2021), it diversifies the coordinates of latent representations within P-Space to calibrate the entangled latent representations. In this context, the use of prototypes as basis vectors defines a structured coordinate system in the P-Space, thereby facilitating the enhancement of generating disentangled representations among samples, as opposed to directly optimizing their representations. To improve computational efficiency, practically, we randomly select 50% of the samples within a minibatch. We posit that this approach can mitigate computational complexity while makes it easier to approximate the model to the optimized state.

**Global Prototype Matrix Orthogonalization Constraint**. Since the P-Space is composed of a set of global prototypes, to better represent the P-Space, these global prototypes should serve as the basis vectors, with each prototype maintaining orthogonal independence from the others. The presence of interdependence among prototypes would invariably compromise the representation efficacy of these prototypes. To ensure the independence of prototypes from one another, the condition of orthogonality must be satisfied. This mandates the following approach:

$$\min \mathcal{L}_{orthogonalization}(\mathcal{B}) = \min(\frac{\|M\|_1}{\|M\|_2^2} + \max(0, |K - \|M\|_1|)), \quad (7)$$

where $M \in [0, 1]^{K \times K}$ and $M_{ij} = \| \cos(\beta_i, \beta_j) \|_1$. The first term $\frac{\|M\|_1}{\|M\|_2^2}$ forces the $M$ to be sparse, i.e., any element $M_{ij} = \| \cos(\beta_i, \beta_j) \|_1$ to be close to 0 ($\beta_i$ and $\beta_j$ is orthogonal) or 1, while the second term motivates $\|M\|_1 \to K$, i.e., only $K$ elements close to 1. Since $M_{ii} = \| \cos(\beta_i, \beta_i) \|_1 = 1, \forall i \in \{1, 2, ..., K\}$, it would force $M_{ij} = \| \cos(\beta_i, \beta_j) \|_1, \forall i, j \in \{1, 2, ..., K\}, i \neq j$ to be 0, i.e. each prototype maintains orthogonal independence from the others.

## 5 EXPERIMENT & ANALYSIS

### 5.1 EXPERIMENT SETUP

**Datasets.** As described before, it is challenging for deep networks to achieve satisfactory performance due to heterogeneity, complexity, and diversity of tabular data. In this paper, we consider a variety of supervised tabular deep learning tasks with heterogeneous features, including binary classification, multiclass classification, and regression. Specifically, the tabular datasets include: Adult (AD) (Kohavi et al., 1996), Higgs (HI) (Vanschoren et al., 2014), Helena (HE) (Guyon et al., 2019), Jannis (JA) (Guyon et al., 2019), ALOI (AL) (Geusebroek et al., 2005), California_housing (CA) (Pace & Barry, 1997). The dataset properties are summarized in Appendix A.1. We split each dataset into training, validation and test set by the ratio of 6:2:2. For the data pre-processing, please refer to Appendix A.1 for more details.

**Baseline Deep Tabular Models.** As the PTARL is a model-agnostic pipeline, we include 6 mainstream deep tabular models to test PTARL's applicability and effectiveness to different predictors with diverse architectures, which are as follows: MLP (Taud & Mas, 2018), ResNet (He et al., 2016), SNN (Klambauer et al., 2017), DCNV2 (Wang et al., 2021), AutoInt (Song et al., 2019), and FT-Transformer (Gorishniy et al., 2021). More details could be found in Appendix A.2.

**PTARL Details.** The PTARL is a two-stage model-agnostic pipeline that aims to enhance the performance of any deep tabular model $F(\cdot; \theta)$ without altering its internal architecture. The first stage is to construct the core of PTARL, i.e. P-Space, that consists of a set of global prototypes $\mathcal{B}$. The number of global prototypes $K$ is data-specific, and we set $K$ to the ceil of $\log(N)$, where $N$ is the total number of features. The estimator $\phi(\cdot; \gamma)$, which is used to calculate the coordinates of representations within P-Space, is a simple 3-layer fully-connected MLP. To ensure fairness, in the second stage of training, we inherit the hyperparameters of $F(\cdot; \theta)$ (the learnable $\theta$ would be re-initialized). We provide the PTARL workflow in Appendix A.3. Following the common practice of previous studies, we use Mean-Square Error (MSE) (lower is better) to evaluate the regression tasks, Accuracy (higher is better) to evaluate binary and multiclass classification tasks. To reduce the effect of randomness, the reported performance is averaged over 10 independent runs.

### 5.2 EMPIRICAL RESULTS

**PTARL generally improves deep tabular models' performance.** From Table 1 we can observe that PTARL achieves consistent improvements over the baseline deep models in all settings. It achieves a more than 4% performance improvement for all settings, whether using Accuracy or RMSE as the evaluation metric. In addition, we conduct Wilcoxon signed-rank test (with $\alpha = 0.05$) (Woolson, 2007) to measure the improvement significance. In all settings, the improvement of PTARL over deep models is statistically significant at the 95% confidence level. This demonstrates the superior adaptability and generality of PTARL to different models and tasks. In addition, the results also indicate that there is no deep model that consistently outperforms others on all tasks, i.e., the universal winner solution does not exist, which is aligned with the findings in previous works (Gorishniy et al., 2021).

**Ablation results.** We further conduct ablation study to demonstrate the effectiveness of key components of PTARL. Specifically, we conduct a comparison between the deep model coupled with PTARL and three variants: (i) PTARL w/o O, that removes the global prototype matrix orthogonalization constraint, (ii) PTARL w/o O, D, that further removes the P-Space coordinates diversifying constraint and (iii) w/o PTARL, that is identical to directly training deep models by Eq. 2. The results in Table 2 show that the removal of any of the components degrades the performance of PTARL. The comparison between PTARL w/o O, D and w/o PTARL indicates that by projecting deep tabular model's representation into P-Space, the shared *global data structure information* are

Table 1: Tabular prediction performance of PTARL over different deep tabular models for different tasks. "↑" represents higher evaluation metric is better for classification, "↓" represents lower evaluation metric is better for regression. The best results are highlighted in bold. "Win" represents the number of datasets that one scheme achieves the best.

|  | MLP | +PTaRL | DCNV2 | +PTaRL | SNN | +PTaRL | ResNet | +PTaRL | AutoInt | +PTaRL | FT-Trans | +PTaRL |
|---|---|---|---|---|---|---|---|---|---|---|---|---|
| AD ↑ | 0.825 | **0.868** | 0.826 | **0.867** | 0.825 | **0.859** | 0.813 | **0.862** | 0.823 | **0.871** | 0.827 | **0.871** |
| HI ↑ | 0.681 | **0.723** | 0.681 | **0.731** | 0.69 | **0.724** | 0.682 | **0.729** | 0.685 | **0.738** | 0.687 | **0.738** |
| HE ↑ | 0.352 | **0.396** | 0.34 | **0.389** | 0.338 | **0.394** | 0.354 | **0.399** | 0.338 | **0.396** | 0.352 | **0.397** |
| JA ↑ | 0.672 | **0.71** | 0.662 | **0.723** | 0.689 | **0.732** | 0.666 | **0.723** | 0.653 | **0.722** | 0.689 | **0.738** |
| AL ↑ | 0.917 | **0.964** | 0.905 | **0.959** | 0.917 | **0.961** | 0.919 | **0.964** | 0.894 | **0.955** | 0.924 | **0.97** |
| CA ↓ | 0.518 | **0.489** | 0.502 | **0.465** | 0.898 | **0.631** | 0.537 | **0.498** | 0.507 | **0.464** | 0.486 | **0.448** |
| Win | 0 | **6** | 0 | **6** | 0 | **6** | 0 | **6** | 0 | **6** | 0 | **6** |

Table 2: Ablation results on the effects of different components in PTARL. The experiment is conducted on FT-Transformer. The best results are highlighted in bold.

|  | AD ↑ | HI ↑ | HE ↑ | JA ↑ | AL ↑ | CA ↓ | Win |
|---|---|---|---|---|---|---|---|
| FT-Transformer + PTARL | **0.871** | **0.738** | **0.397** | **0.738** | **0.97** | **0.448** | 6 |
| PTARL w/o O | 0.859 | 0.722 | 0.383 | 0.725 | 0.96 | 0.453 | 0 |
| PTARL w/o O, D | 0.841 | 0.704 | 0.369 | 0.702 | 0.94 | 0.466 | 0 |
| w/o PTARL | 0.827 | 0.687 | 0.352 | 0.689 | 0.924 | 0.486 | 0 |

captured by global prototypes to solve the *representation localization* to enhance the representation quality. In addition, diversifying the representation coordinates in P-Space and orthogonalizing the global prototypes of P-Space could both enable the generation of *disentangled representations* to alleviate the *representation entanglement* problem. Besides, we also provide additional ablation tests, including validating the effectiveness of K-Means as the global prototypes initialization method and the effectiveness of Optimal Transport (OT) as the distribution measurement in Table 3. The result of initialization method is aligned to our explanation for the advantage of K-Means in Section 4.2. In addition, the result of distribution measurement indicates that compared to Manhattan distance and Euclidean distance, OT measures the minimum distance between two distributions through point by point calculation, which could better capture the data structure between two distributions. Due to the space limit, we leave more details and results about the above ablation study to Appendix A.4.

**Computational efficiency and sensitivity analysis.** In our paper, while Optimal Transport (OT) demonstrates strong computational capabilities for measuring the minimum distance between two distributions, it does not significantly increase computational complexity. We include the computational efficiency details in Appendix A.5. In addition, we also incorporate the sensitivity analysis for the weights of different loss functions in Appendix A.6. Since the core of PTARL is the constructed P-Space, we further explore PTARL's performance under different global prototypes number in Appendix A.6.

## 5.3 PTARL PROCEDURE ANALYSIS AND VISUALIZATION

**PTARL enables the generation of disentangled representation.** Fig. 3 shows the learned representation of deep model $F(\cdot; \theta)$ w/ and w/o PTARL on binary and multiclass classification tasks. The first row shows that different deep models suffer from the *representation entanglement* problem and this is aligned with our motivation. With PTARL, representations within P-Space are separated into several disjoint regions, where representations with similar labels are in the same regions. This

Table 3: Ablation results on the effects of different prototype initialization and distribution measurement in PTARL. The experiment is conducted on FT-Transformer. The best results are highlighted in bold.

| Initialization | AD ↑ | HI ↑ | HE ↑ | JA ↑ | AL ↑ | CA ↓ |
|---|---|---|---|---|---|---|
| PTARL w/ K-Means (Ours) | **0.871** | **0.738** | **0.397** | **0.738** | **0.97** | **0.448** |
| PTARL w/ Random | 0.863 | 0.721 | 0.389 | 0.729 | 0.951 | 0.462 |

| Distribution measurement | AD ↑ | HI ↑ | HE ↑ | JA ↑ | AL ↑ | CA ↓ |
|---|---|---|---|---|---|---|
| PTARL w/ OT (Ours) | **0.871** | **0.738** | **0.397** | **0.738** | **0.97** | **0.448** |
| PTARL w/ Manhattan distance | 0.859 | 0.729 | 0.382 | 0.721 | 0.953 | 0.454 |
| PTARL w/ Euclidean distance | 0.862 | 0.713 | 0.390 | 0.724 | 0.949 | 0.459 |

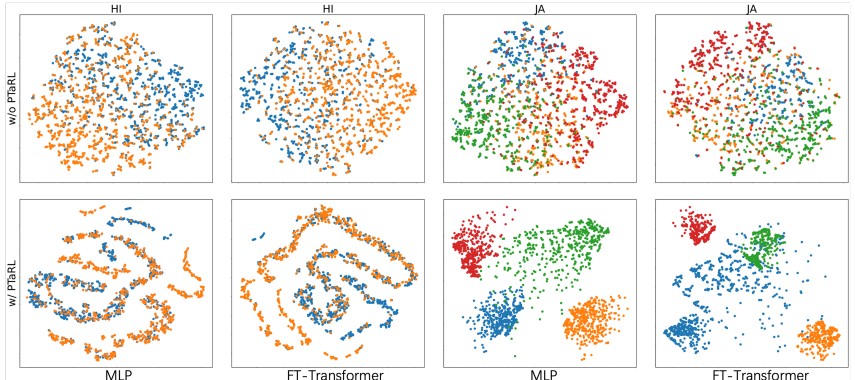

Figure 3: Visualization of learned representations of deep tabular models w/ and w/o PTARL.

demonstrates PTARL' ability to generate *disentangled representations* for tabular deep learning.

**PTARL enables P-Space coordinates diversifying.** To validate whether the learned coordinates of representation within P-Space have been diversified, we average the coordinates of data points which are belong to the same category and visualize them in Fig. 4. Without the Diversifying Constraint, coordinates of different categories appear similar, but incorporating the constraint enhances the diversity between category coordinates. **PTARL enables the orthogonalization of global prototypes within P-Space.** To explore the structure of constructed P-Space, we visualize the relation between any $(\beta_i, \beta_j)$ by calculating $\|\cos(\beta_i, \beta_j)\|_1$ in Fig. 5. Each prototype maintains orthogonal independence from the others, which demonstrates the effectiveness of prototype matrix orthogonalization constraint.

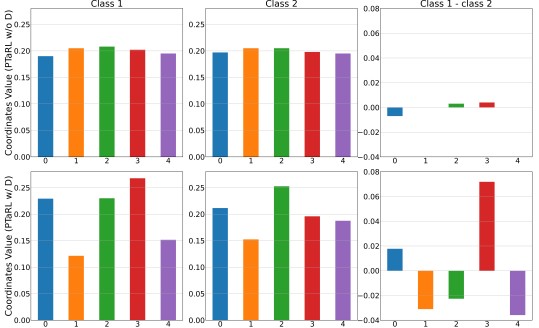

Figure 4: P-Space coordinates diversifying visualization of FT-Transformer on HI w/o and w/ Coordinates Diversifying Constraint (D). The first column and second column correspond to the average coordinates values of two different categories, the third column represents the difference of the two categories.

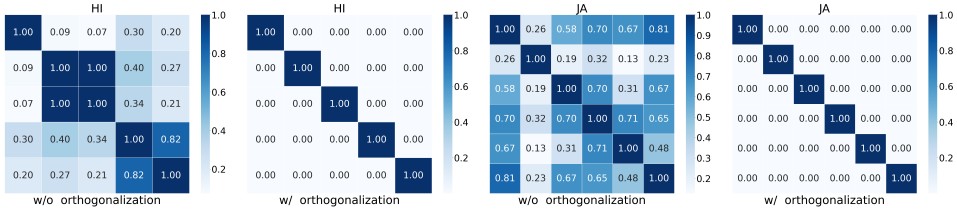

Figure 5: Global prototypes orthogonalization visualization of MLP on two different tasks.

## 6 CONCLUSION

In this paper, we have investigated the learned patterns of deep tabular models and identified two inherent representation challenges hindering satisfactory performance, i.e. *sample localization* and *representation entanglement*. To handle these challenges, we proposed PTARL, a prototype-based tabular representation learning pipeline that can be coupled with any deep tabular model to enhance the representation quality. The core of the PTARL is the constructed P-Space, that consists of a set of global prototypes. PTARL mainly involves two stages, i.e. global prototype generation and projecting representations into P-Space, to capture the *global data structure information*. Besides, two constraints are designed to *disentangle* the projected representations within P-Space. The empirical results on various real world tasks demonstrated the effectiveness of PTARL for tabular deep learning. Our work can shed some light on developing better algorithms for similar tasks.

ACKNOWLEDGMENTS

This work is supported by the National Natural Science Foundation of China (No.61976102, No.U19A2065, No.62306125) and Key R&D Program of the Ministry of Science and Technology, China (2023YFF0905400).

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

# A    APPENDIX

In this section, we provide details of the datasets, baseline methods, the implementation of our method, comprehensive experimental results and visualizations. More detailed information is available at `https://github.com/HangtingYe/PTaRL`.

## A.1    DATASETS DETAILS.

**Data pre-processing.** Due to the property of neural networks, data pre-processing is important, especially for tabular data. To handle categorical features, we adopt an integer encoding scheme, where each category within a column is uniquely mapped to an integer to index the embedding in lookup table. Furthermore, we maintain consistent embedding dimensions for all categorical features. For numerical features, we apply column-wise normalization method. In regression tasks, we also apply the normalization to the labels. To ensure fair comparisons, we adhere to identical preprocessing procedures across all deep networks for each dataset. Following (Gorishniy et al., 2021), we use the quantile transformation from the Scikit-learn library (Pedregosa et al., 2011). We apply Standardization to HE and AL. The latter one represents image data, and standardization is a common practice in computer vision.

Table 4: Tabular data properties. Accuracy is used for binary and multiclass classification, RMSE denotes Root Mean Square Error for regression.

|                        | AD       | HI       | HE       | JA       | AL       | CA    |
|------------------------|----------|----------|----------|----------|----------|-------|
| Objects                | 48842    | 98050    | 65196    | 83733    | 108000   | 20640 |
| Numerical, Categorical | 6, 8     | 28, 0    | 27, 0    | 54, 0    | 128, 0   | 8, 0  |
| Classes                | 2        | 2        | 100      | 4        | 1000     | -     |
| metric                 | Accuracy | Accuracy | Accuracy | Accuracy | Accuracy | RMSE  |

## A.2    BASELINE DEEP TABULAR MODELS DETAILS.

- MLP (Taud & Mas, 2018).

- DCNV2 (Wang et al., 2021). Consists of an MLP-like module and the feature crossing module (a combination of linear layers and multiplications).

- SNN (Klambauer et al., 2017). An MLP-like architecture with the SELU activation that enables training deeper models.

- ResNet (He et al., 2016). The key innovation is the use of residual connections, also known as skip connections or shortcut connections. These connections enable the network to effectively train very deep neural networks, which was challenging before due to the vanishing gradient problem. In this paper, we use the ResNet version introduced by (Gorishniy et al., 2021).

- AutoInt (Song et al., 2019). Transforms features to embeddings and applies a series of attention-based transformations to the embeddings.

- FT-Transformer (Gorishniy et al., 2021). FT-Transformer is introduced by (Gorishniy et al., 2021) to further improved AutoInt through better token embeddings.

## A.3    PTARL WORKFLOW.

---

**Algorithm 1** PTARL algorithm workflow.

---

**Input:** Input data $D = \{X, Y\}$, deep tabular model $F(\cdot; \theta) = G_h(G_f(\cdot; \theta_f); \theta_h)$, coordinates estimator $\phi(\cdot; \gamma)$

1: **Phase 1: prototype generating**
2: Train $F(\cdot; \theta)$ by $\min \mathcal{L}_{task}(X, Y) = \min_{\theta_f, \theta_h} \mathcal{L}(G_h(G_f(X; \theta_f); \theta_h), Y)$ (Eq.2)
3: Obtain global prototypes $\mathcal{B} = \{\beta_k\}_{k=1}^{K} \in \mathbb{R}^{K \times d}$ through applying K-Means clustering to the output of the trained backbone $G_f(X; \theta_f)$
4: **Phase 2: representation projecting**
5: Re-initialize the parameters of $F(\cdot; \theta) = G_h(G_f(\cdot; \theta_f); \theta_h)$
6: **while** $\mathcal{B}, \theta_f, \theta_h, \gamma$ has not converged **do**
7:     Sample minibatch of size $n_b$ from $D$
8:     **for** $i \leftarrow 1$ to $n_b$ **do**
9:         Obtain $i$-th instance representation distribution by $P_i = \delta_{G_f(x_i; \theta_f)}$
10:         Calculate $i$-th instance projection representation coordinates $r_i = \phi(G_f(x_i; \theta_f); \gamma)$
11:         Obtain $i$-th instance projection representation distribution: $Q_i = \sum_{k=1}^{K} r_i^k \delta_{\beta_k}$
12:         Calculate the OT distance between $P_i$ and $Q_i$: $\text{OT}(P_i, Q_i) = \min_{\mathbf{T}_i \in \Pi(P_i, Q_i)} \langle \mathbf{T}_i, \mathbf{C}_i \rangle$
13:     **end for**
14:     Average the OT distance within the minibatch to compute $\mathcal{L}_{projecting}(X, \mathcal{B}) = \frac{1}{n_b} \sum_{i=1}^{n_b} \text{OT}(P_i, Q_i) = \frac{1}{n_b} \sum_{i=1}^{n_b} \min_{\mathbf{T}_i \in \Pi(P_i, Q_i)} \langle \mathbf{T}_i, \mathbf{C}_i \rangle$ (Eq.4)
15:     Compute $\mathcal{L}_{task}(X, Y) = \frac{1}{n_b} \sum_{i=1}^{n_b} \mathcal{L}(G_h(\sum_{k=1}^{K} r_i^k \beta_k; \theta_h), y_i)$ (Eq.5)
16:     Random select 50% of the samples within minibatch to compute $\mathcal{L}_{diversifying}(X) = -\sum_{i=1}^{n_b} \sum_{j=1}^{n_b} \mathbf{1}\{y_i, y_j \in \text{positive pair}\} \log \frac{\exp(\cos(r_i, r_j))}{\sum_{i=1}^{n_b} \sum_{j=1}^{n_b} \exp(\cos(r_i, r_j))}$ (Eq.6)
17:     Compute $\mathcal{L}_{orthogonalization}(\mathcal{B}) = (\frac{\|M\|_1}{\|M\|_2^2} + \max(0, |K - \|M\|_1|))$ (Eq.7)
18:     Update $\mathcal{B}, \theta_f, \theta_h, \gamma$ by minimizing $\mathcal{O} = \mathcal{L}_{task} + \mathcal{L}_{projecting} + \mathcal{L}_{diversifying} + \mathcal{L}_{orthogonalization}$ through gradient descent
19: **end while**
20: **return** $G_f, G_h, \phi, \mathcal{B}$

---

## A.4 COMPARISON WITH BASELINE DEEP NETWORKS.

Table 5: Ablation results on the effects of different components in PTARL.

|  | AD ↑ | HI ↑ | HE ↑ | JA ↑ | AL ↑ | CA ↓ | Win |
|---|---|---|---|---|---|---|---|
| MLP + PTARL | **0.868** | **0.723** | **0.396** | **0.71** | **0.964** | **0.489** | 6 |
| PTARL w/o O | 0.856 | 0.711 | 0.381 | 0.703 | 0.955 | 0.496 | 0 |
| PTARL w/o O, D | 0.838 | 0.696 | 0.364 | 0.683 | 0.937 | 0.511 | 0 |
| w/o PTARL | 0.825 | 0.681 | 0.352 | 0.672 | 0.917 | 0.518 | 0 |
| ResNet + PTARL | **0.862** | **0.729** | **0.399** | **0.723** | **0.964** | 0.498 | 5 |
| PTARL w/o O | 0.848 | 0.712 | 0.383 | 0.707 | 0.949 | **0.493** | 1 |
| PTARL w/o O, D | 0.83 | 0.698 | 0.368 | 0.69 | 0.933 | 0.518 | 0 |
| w/o PTARL | 0.813 | 0.682 | 0.354 | 0.666 | 0.919 | 0.537 | 0 |
| FT-Transformer + PTARL | **0.871** | **0.738** | **0.397** | **0.738** | **0.97** | **0.448** | 6 |
| PTARL w/o O | 0.859 | 0.722 | 0.383 | 0.725 | 0.96 | 0.453 | 0 |
| PTARL w/o O, D | 0.841 | 0.704 | 0.369 | 0.702 | 0.94 | 0.466 | 0 |
| w/o PTARL | 0.827 | 0.687 | 0.352 | 0.689 | 0.924 | 0.486 | 0 |

## A.5 COMPUTATIONAL EFFICIENCY DETAILS.

To approximate the optimal transport (OT) distance between two discrete distributions of size $n$, the time complexity bound scales as $n^2 \log(n)/\epsilon^2$ to reach $\epsilon$-accuracy with Sinkhorn's algorithm, as demonstrated by (Chizat et al., 2020; Altschuler et al., 2017). In this paper, for each instance $x_i$, we pushing $G_f(x_i; \theta_f)$'s representation distribution $P_i$ to the corresponding projection representation distribution $Q_i$ by minimizing their OT distance. Thus the instance-wise time complexity bound scales as $O(K^2 \log(K)/\epsilon^2)$, where $K$ is set to the number of global prototypes. The number of global prototypes $K$ is data-specific, and we set $K$ to the ceil of $\log(N)$, where $N$ is the total number of features. While OT demonstrates strong computational capabilities for measuring the minimum distance between two distributions, it does not significantly increase computational complexity in our setting.

We also compare the computational cost of PTARL on a single GTX 3090 GPU. We report the computational cost (s) of PTARL per training epoch on different datasets. As shown in Table 6, coupling deep models with PTARL produces a better performance on all datasets with an acceptable cost.

Table 6: Computational cost (s) per training epoch for PTARL.

|  | AD | HI | HE | JA | AL | CA |
|---|---|---|---|---|---|---|
| MLP + PTARL phase1 | 0.230 | 0.482 | 0.312 | 0.402 | 0.520 | 0.107 |
| MLP + PTARL phase2 | 1.030 | 2.435 | 0.909 | 1.204 | 1.469 | 0.543 |
| DCNV2 + PTARL phase1 | 0.436 | 1.000 | 0.653 | 0.840 | 1.081 | 0.210 |
| DCNV2 + PTARL phase2 | 1.217 | 2.831 | 1.159 | 1.485 | 1.934 | 0.599 |
| SNN + PTARL phase1 | 0.257 | 0.506 | 0.343 | 0.446 | 0.555 | 0.117 |
| SNN + PTARL phase2 | 1.025 | 2.371 | 0.926 | 1.162 | 1.528 | 0.533 |
| ResNet + PTARL phase1 | 0.413 | 0.919 | 0.616 | 0.783 | 1.019 | 0.196 |
| ResNet + PTARL phase2 | 1.131 | 2.697 | 1.167 | 1.451 | 1.908 | 0.568 |
| AutoInt + PTARL phase1 | 0.655 | 2.278 | 1.394 | 3.575 | 7.278 | 0.304 |
| AutoInt + PTARL phase2 | 2.009 | 9.976 | 5.578 | 22.884 | 23.444 | 0.707 |
| FT-Transformer + PTARL phase1 | 0.828 | 2.367 | 1.489 | 3.130 | 9.194 | 0.418 |
| FT-Transformer + PTARL phase2 | 1.579 | 4.176 | 1.976 | 3.798 | 9.962 | 0.804 |

A.6    SENSITIVITY ANALYSIS.

**The influence of loss weights.** Altogether, the proposed PTARL aims to minimize the following objective function in stage 2:

$$\mathcal{O} = \mathcal{L}_{task}(X, Y) + \mathcal{L}_{projecting}(X, \mathcal{B}) + \mathcal{L}_{diversifying}(X) + \mathcal{L}_{orthogonalization}(\mathcal{B}). \quad (8)$$

The weights of above losses are set to 1.0, 0.25, 0.25 and 0.25 respectively and the weights are fixed in all settings. We provide the sensitivity analysis for the loss weights conducted on FT-Transformer in Table 7. The results indicate that PTARL is robust to the loss weights.

Table 7: Sensitivity analysis for loss weights.

| Loss weights | AD ↑ | HI ↑ | HE ↑ | JA ↑ | AL ↑ | CA ↓ |
|---|---|---|---|---|---|---|
| **1.0, 0.25, 0.25, 0.25** | 0.871 | 0.738 | 0.397 | 0.738 | 0.97 | 0.448 |
| 1.0, 0.25, 0.25, 0.2 | 0.871 | 0.740 | 0.398 | 0.737 | 0.97 | 0.449 |
| 1.0, 0.25, 0.25, 0.15 | 0.870 | 0.739 | 0.397 | 0.737 | 0.97 | 0.448 |
| 1.0, 0.25, 0.25, 0.1 | 0.869 | 0.737 | 0.395 | 0.736 | 0.969 | 0.450 |
| 1.0, 0.25, 0.25, 0.05 | 0.864 | 0.734 | 0.391 | 0.731 | 0.964 | 0.451 |
| 1.0, 0.25, 0.25, 0.0 | 0.859 | 0.722 | 0.383 | 0.725 | 0.96 | 0.453 |
| **1.0, 0.25, 0.25, 0.25** | 0.871 | 0.738 | 0.397 | 0.738 | 0.97 | 0.448 |
| 1.0, 0.25, 0.15, 0.25 | 0.869 | 0.737 | 0.397 | 0.736 | 0.969 | 0.451 |
| 1.0, 0.25, 0.05, 0.25 | 0.869 | 0.737 | 0.395 | 0.734 | 0.969 | 0.452 |
| 1.0, 0.25, 0.0, 0.25 | 0.861 | 0.724 | 0.379 | 0.722 | 0.961 | 0.456 |

**The influence of global prototype number to PTARL.** Since the core of PTARL is the constructed P-Space, we further explore PTARL's performance under different global prototypes number. Fig. 6 shows the results of average performance of PTARL and its variants over all deep models. The PTARL 's performance gradually improves with global prototypes number increasing, and finally reaches a stable level. Thus it is reasonable to set the number of prototypes $K$ as the ceil of $\log(N)$, where $N$ is the total number of features.

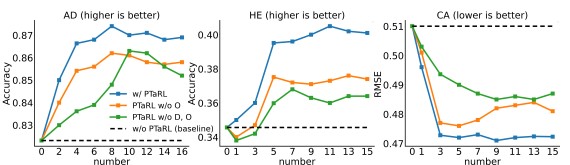

Figure 6: Tabular prediction performance of the variants of PTARL with various numbers of prototypes. The results are averaged over all baseline deep tabular models under binary classification (AD), multiclass classification (HE) and regression (CA).

A.7    ADDITIONAL RESULTS.

Table 8: The influence of DNN depths to the performance of PTARL.

| Datasets | Models | layers = 3 | layers = 5 | layers = 7 | layers = 9 |
|---|---|---|---|---|---|
| HI ↑ | MLP | 0.680 | 0.707 | 0.675 | 0.669 |
| | MLP + PTARL | **0.719** | **0.730** | **0.729** | **0.732** |
| | FT-Transformer | 0.687 | 0.719 | 0.709 | 0.691 |
| | FT-Transformer + PTARL | **0.738** | **0.742** | **0.741** | **0.737** |
| JA ↑ | MLP | 0.670 | 0.715 | 0.704 | 0.689 |
| | MLP + PTARL | **0.708** | **0.732** | **0.728** | **0.733** |
| | FT-Transformer | 0.689 | 0.716 | 0.729 | 0.709 |
| | FT-Transformer + PTARL | **0.738** | **0.741** | **0.745** | **0.742** |
| CA ↓ | MLP | 0.522 | 0.513 | 0.509 | 0.524 |
| | MLP + PTARL | **0.491** | **0.480** | **0.482** | **0.479** |
| | FT-Transformer | 0.486 | 0.476 | 0.472 | 0.478 |
| | FT-Transformer + PTARL | **0.448** | **0.446** | **0.442** | **0.443** |

A.8 DETAILED EXPLANATION OF THE OPTIMIZATION PROCESS OF EQ. 4.

The Optimal Transport (OT) problem is usually to find the most cost-effective way to transform a given distribution to another distribution, which is typically achieved by calculating the specified transportation plan that minimizes the total transportation cost, while the minimized cost is usually called OT distance. In our paper, we minimize the distance between original representation distribution over each sample $P_i$ by deep tabular models and the corresponding projection representation distribution $Q_i$ in P-Space with global prototypes, in order to preserve original data information (of $P_i$) in $Q_i$. We follow the typical setting of OT problem to first estimate the transport plan to obtain the OT distance between $P_i$ and $Q_i$. Then, the obtained OT distance is further used as loss function to jointly learn the two representations.

Specifically, after initializing the global prototypes $\mathcal{B}$ of P-Space, we project the original data samples into P-Space to learn the representations with global data structure information. To better illustrate the optimization process, we revise the Eq. 4 in the original paper to make it more readable. In Eq. 4, the $i$-th sample representation by deep tabular model is denoted as $G_f(x_i; \theta_f)$, the empirical distribution over this sample representation is $P_i = \delta_{G_f(x_i; \theta_f)}$, the projection representation distribution is denoted as: $Q_i = \sum_{k=1}^{K} r_i^k \delta_{\beta_k}$, where $r_i$ is coordinates. To capture the shared global data structure information, we formulate the representation projecting as the process of extracting instance-wise data information by $G_f(x_i; \theta_f)$ to $P_i$, and then pushing $P_i$ towards $Q_i$ to encourage each prototype $\beta_k$ to capture the shared global data structure information, a process achieved by minimizing the OT distance between $P_i$ and $Q_i$. The OT distance between $P_i$ and $Q_i$ could first be calculated by: $\text{OT}(P_i, Q_i) = \min_{\mathbf{T}_i \in \Pi(P_i, Q_i)} \langle \mathbf{T}_i, \mathbf{C}_i \rangle$, where $C_{ik} = 1 - \cos(G_f(x_i; \theta_f), \beta_k)$, the average OT distance between $P_i$ and $Q_i$ over train sets could be viewed as loss function $\mathcal{L}_{projecting}(X, \mathcal{B})$ to be further optimized: $\min \mathcal{L}_{projecting}(X, \mathcal{B}) = \min \frac{1}{n} \sum_{i=1}^{n} \text{OT}(P_i, Q_i) = \min_{\theta_f, \gamma, \mathcal{B}} \frac{1}{n} \sum_{i=1}^{n} \min_{\mathbf{T}_i \in \Pi(P_i, Q_i)} \langle \mathbf{T}_i, \mathbf{C}_i \rangle$, we use gradient descent to update $\theta_f, \gamma, \mathcal{B}$.

A.9 DETAILED DESCRIPTION OF THE *global data structure information* AND *sample localization*.

In the context of any tabular dataset, we have observed *global data structure information* comprises two different components: (i) the global feature structure and (ii) the global sample structure.

Considering the feature structure, traditional and deep tabular machine learning methods utilize all features or a subset of features as input, allowing them to model inherent interactions among features and thereby acquire a comprehensive global feature structure. In addition, there also exists the sample structure given a tabular dataset. Traditional methods (e.g., boosted trees) can effectively model the overarching relationships between data samples. Specifically, in XGBoost, the dataset undergoes partitioning by comparing all the samples, with each node of a decision tree representing a specific partition, and each leaf node corresponding to a predictive value. The iterative splitting of nodes during training empowers decision trees in XGBoost to learn the distribution of all the samples across distinct regions of the data space, capturing global sample structure.

However, we note that deep tabular machine learning methods typically rely on batch training to obtain data representations within a batch. These methods do not explicitly consider the structure between samples within a batch. Furthermore, they fail to capture the global structure between samples across different batches. This limitation presents challenges in comprehensively capturing global data distribution information, consequently impeding overall performance.

Our methods rebuild the representation space with global prototypes (P-Space) in the first stage. Then in the second stage, the original data representation by deep tabular machine learning methods is projected into P-Space to obtain projection representation with global prototypes. On the one hand, by minimizing the Optimal Transport distance between the two representations, we could represent each sample with global prototypes, and in the meanwhile encode the global feature structure in learning global prototypes, considering backbone models can inherently learn the interactions among features. On the other hand, the global prototypes are learned by directly modeling all the data samples and thus the complex data distribution could be obtained by global prototypes to capture the global sample structure. Therefore, PTARL is able to capture both the feature and sample structure information by prototype learning. Considering previous deep tabular machine learning

methods can only acquire the representations limited by the batch training, we use the concept of *sample localization* to encapsulate this limitation.

### A.10 GLOBAL PROTOTYPES ORTHOGONALIZATION VISUALIZATION.

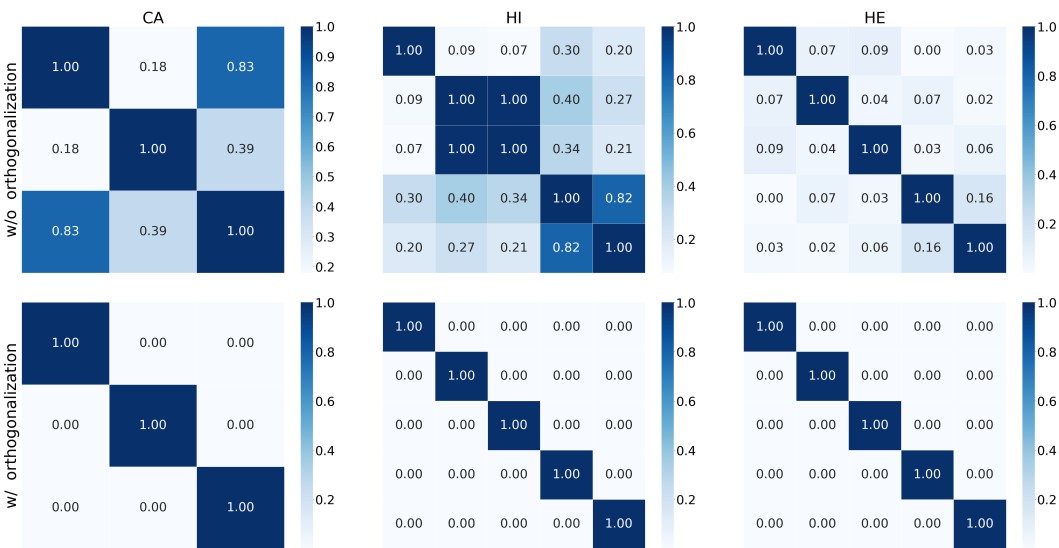

Figure 7: Global prototypes orthogonalization visualization of MLP.

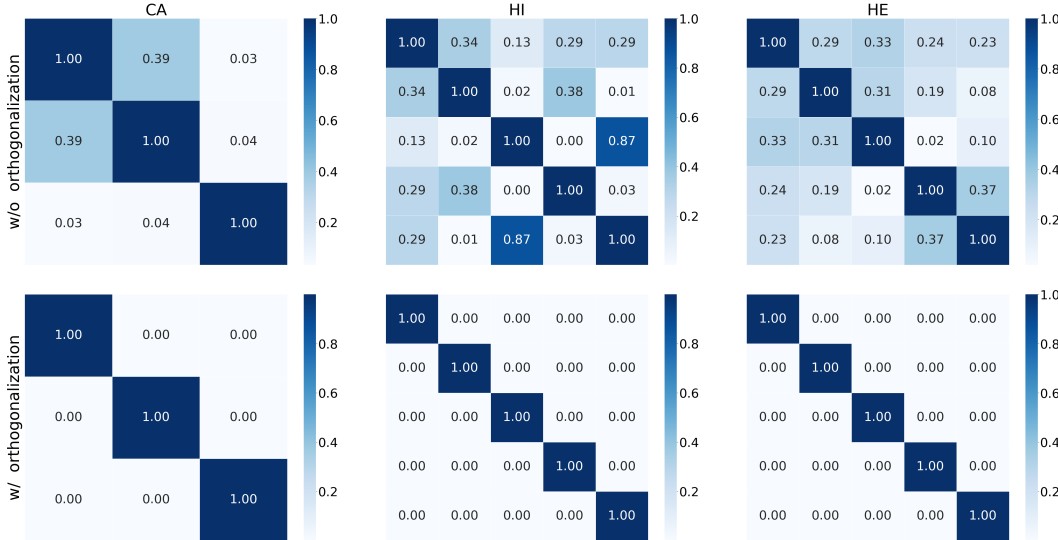

Figure 8: Global prototypes orthogonalization visualization of ResNet.

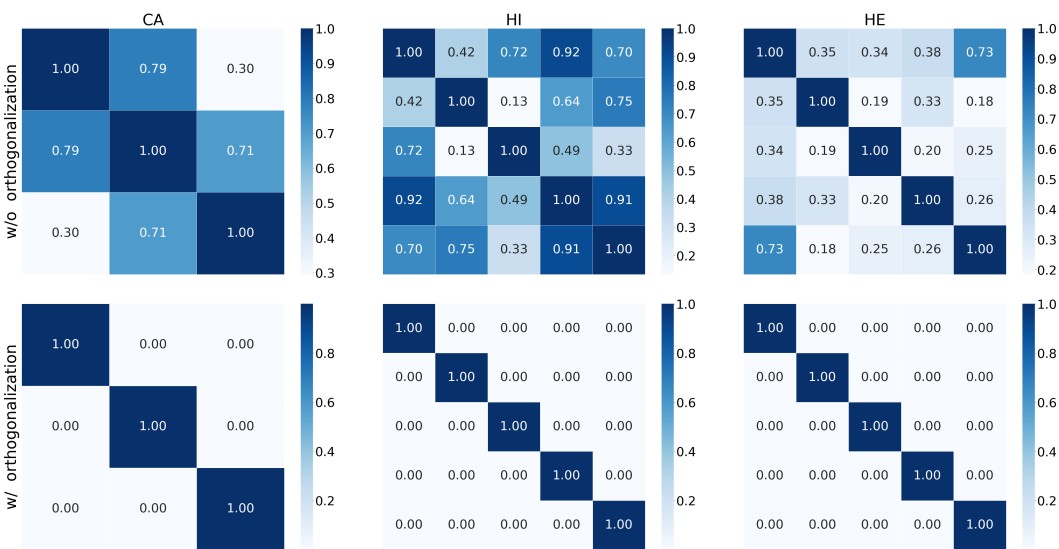

Figure 9: Global prototypes orthogonalization visualization of FT-Transformer.

