# OpenReview forum: "PTaRL: Prototype-based Tabular Representation Learning via Space Calibration"
_ICLR.cc/2024/Conference — ICLR 2024 spotlight_

### Official Review · Reviewer_GXco · 2023-10-31

**Soundness:** 3 good
**Presentation:** 3 good
**Contribution:** 2 fair
**Rating:** 8
**Confidence:** 3

**Summary:**

This paper presents PTaRL, a model-agnostic method to enhance deep-learning methods for tabular data prediction. The method inserts a sound prototype learning step after the penultimate layer of any DNN to alleviate the issue of representation entanglement and localization. The results show improvement across several architectures and datasets.

**Strengths:**

- The problems identified (representation entanglement and localization) and the results (including the ablation study) are convincing.
- The various steps — the joint optimization of DNN representation, prototypes, and projection coordinate plus the two constraints — are intuitive.
- The paper reads smoothly.

**Weaknesses:**

- The concepts of “global data structure information” and “sample location” are not very clear, at least not as concretely demonstrated as entanglement and orthogonality.

**Questions:**

The whole purpose of using the prototype seems to be for capturing “global data structure information” so as to avoid “sample localization”. However, after reading Section 4, I am still unclear what “global data structure information” really is. Could the authors provide a more explicit definition and description of it? Similarly for “sample location.”

I feel Figure 4 is a much better example of disentanglement than Figure 1, because I still see substantial overlap in the bottom row of Figure 1.

I don’t quite understand how Figure 5 shows diversification.

I was wondering where the boosted performance sits in the literature as a whole. The paper shows DNN and DNN + PTaRL. Do these now match the performance of XGBoost and other state-of-the-art tree-based methods? How much better are they compared to older methods, such as kernel prototype classification and regression? Is it possible to apply PTaRL on different DNN depths to show the contribution of deep-learning representation vs the contribution of the prototype representation? I think knowing the answer to the first question (related to XGBoost) will be very useful. I can understand if the authors feel the other questions are more distracting than useful, since the paper has a focus on enhancing deep-learning approaches.

Overall, the paper is quite well rounded. The problems, solutions, implementations, and results all work together well.

---

> ### Author Response · Authors · 2023-11-21
> **Response (Part 1/3)**
>
> Thanks for your comments. We respond to your questions as follows and hope they can address your concerns.
>
>
> **Q1**: Unclear definition of the concepts of *global data structure information* and *sample localization*.
>
> **R**:
> In the context of any tabular dataset, we have observed *global data structure information* comprises two different components: (i) the global feature structure and (ii) the global sample structure.
>
> Considering the feature structure, traditional and deep tabular machine learning methods utilize all features or a subset of features as input, allowing them to model inherent interactions among features and thereby acquire a comprehensive global feature structure.
> In addition, there also exists the sample structure given a tabular dataset. Traditional methods (e.g., boosted trees) can effectively model the overarching relationships between data samples. Specifically, in XGBoost, the dataset undergoes partitioning by comparing all the samples, with each node of a decision tree representing a specific partition, and each leaf node corresponding to a predictive value. The iterative splitting of nodes during training empowers decision trees in XGBoost to learn the distribution of all the samples across distinct regions of the data space, capturing global sample structure.
>
>
> However, we note that deep tabular machine learning methods typically rely on batch training to obtain data representations within a batch. These methods do not explicitly consider the structure between samples within a batch.
> Furthermore, they fail to capture the global structure between samples across different batches. This limitation presents challenges in comprehensively capturing global data distribution information, consequently impeding overall performance.
>
> Our methods rebuild the representation space with global prototypes (P-Space) in the first stage.
> Then in the second stage, the original data representation by deep tabular machine learning methods is projected into P-Space to obtain projection representation with global prototypes.
> On the one hand, by minimizing the Optimal Transport distance between the two representations, we could represent each sample with global prototypes, and in the meanwhile encode the global feature structure in learning global prototypes, considering backbone models can inherently learn the interactions among features.
> On the other hand, the global prototypes are learned by directly modeling all the data samples and thus the complex data distribution could be obtained by global prototypes to capture the global sample structure.
> Therefore, PTaRL is able to capture both the feature and sample structure information by prototype learning. Considering previous deep tabular machine learning methods can only acquire the representations limited by the batch training, we use the concept of *sample localization* to encapsulate this limitation. We also provide the detailed explanation in Appendix A.8.
>
> **Q2**: Doubts about the Fig. 1 and Fig. 4.
>
> **R**:
> Thanks for your mention.
> We also note that there exists certain overlap in Fig. 1.
> After our careful analysis, we attribute the differences between Fig. 1 and Fig. 4 to the different characteristics of distinct datasets.
>
> Specifically, Fig. 1 is based on the Adult dataset, while Fig. 4 is derived from the Higgs and Jannis datasets.
> In comparison to the Higgs dataset (98050 samples, 28 features) and the Jannis dataset (83733 samples, 54 features), the Adult dataset (48842 samples, 14 features) has a smaller number of samples and features.
> Additionally, the Adult dataset includes both numerical and categorical features, whereas the Higgs and Jannis datasets only contain numerical features.
> Considering the smaller size of the Adult dataset for training and its more intricate feature relationships, it poses a more challenging task for deep tabular machine learning methods to learn disentangled representations.
> Our proposed PTaRL offers a viable solution to enhance the disentanglement of representations by deep tabular machine learning methods.
>
>
>
> **Q3**: Unclear description of the diversification in Fig. 5.
>
> **R**: We apologize for any confusion and have enhanced the clarity of Fig. 5 for improved illustration.
> The top and bottom rows respectively depict scenarios without and with the Diversifying Constraint.
> The first and second columns represent the mean coordinate values of two distinct categories, while the third column illustrates the difference between the two categories.
> The first row shows that without Diversifying Constraint, the average coordinates between two different categories are similar, posing challenges in effectively distinguishing representations belonging to distinct categories.
> Contrastingly, the second row, which introduces the Diversifying Constraint, showcases a more pronounced difference in coordinates between categories, thereby promoting diversification in the coordinates of distinct categories.

---

> > ### Author Response · Authors · 2023-11-21
> > **Response (Part 2/3)**
> >
> > **Q4.1**: Comparison between PTaRL and state-of-the-art tree-based methods.
> >
> > **R**: Thanks for valuable suggestion. We have conducted a comparison of the performance of PTaRL (utilizing FT-Transformer as a backbone) with two state-of-the-art tree-based methods, namely XGBoost [1] and LightGBM [2].
> > The experimental results are detailed below and can also be referenced in Table 9 in the Appendix.
> > $\uparrow$ represents higher evaluation metric is better for classification and $\downarrow$ represents lower evaluation metric is better for regression.
> > In comparison to XGBoost, PTaRL performs better across all experimental settings.
> > When contrasted with LightGBM, PTaRL outperforms it in 4 out of 6 settings, with a slight performance dip observed in the remaining 2 settings.
> > In summary, PTaRL consistently matches the performance of state-of-the-art tree-based methods across various settings.
> > The utilization of deep learning for tabular data, allowing the construction of multi-model pipelines and knowledge transfer across different domains, is an appealing prospect. Our proposed method holds the potential to enhance the application of deep learning in the tabular domain.
> >
> >
> > |                                 | AD $\uparrow$  | HI $\uparrow$  | HE $\uparrow$  | JA $\uparrow$  | AL $\uparrow$ | CA $\downarrow$ |
> > |---------------------------------|----------------|----------------|----------------|----------------|---------------|-----------------|
> > | XGBoost                         | 0.856          | 0.691          | 0.361          | 0.724          | 0.954         | 0.462           |
> > | LightGBM                        | 0.860          | 0.713          | 0.382          | **0.743**      | 0.968         | **0.445**       |
> > | FT-Transformer + PTaRL          | **0.871**      | **0.738**      | **0.397**      | 0.738          | **0.97**      | 0.448           |
> >
> > [1] Xgboost: A scalable tree boosting system (Chen, Tianqi and Guestrin, Carlos) [KDD 2016]
> >
> > [2] Lightgbm: A highly efficient gradient boosting decision tree (Ke, Guolin and Meng, Qi and Finley, Thomas and Wang, Taifeng and Chen, Wei and Ma, Weidong and Ye, Qiwei and Liu, Tie-Yan) [NeurIPS 2017]
> >
> > **Q4.2**: Comparison between PTaRL and other classical machine learning methods.
> >
> > **R**: Thanks for constructive suggestion.
> > We conducted a performance comparison between PTaRL (with FT-Transformer as backbone) and classical machine learning methods, namely Support Vector Machine (SVM) [3] for classification, Support Vector Regression (SVR) for regression, and K-Nearest Neighbors (KNN) [4].
> > The detailed experimental results are provided below and can also be referenced in Table 10 in the Appendix.
> > $\uparrow$ represents higher evaluation metric is better for classification and $\downarrow$ represents lower evaluation metric is better for regression.
> > The results show that PTaRL performs significantly better than SVM, SVR, and KNN.
> >
> >
> >
> > |                                 | AD $\uparrow$  | HI $\uparrow$  | HE $\uparrow$  | JA $\uparrow$  | AL $\uparrow$ | CA $\downarrow$ |
> > |---------------------------------|----------------|----------------|----------------|----------------|---------------|-----------------|
> > | SVM (SVR for regression)        | 0.823          | 0.683          | 0.340          | 0.658          | 0.901         | 0.515           |
> > | KNN                             | 0.816          | 0.680          | 0.345          | 0.652          | 0.889         | 0.522           |
> > | FT-Transformer + PTaRL          | **0.871**      | **0.738**      | **0.397**      | **0.738**      | **0.97**      | **0.448**       |
> >
> > [3] LIBSVM: a library for support vector machines (Chang, Chih-Chung and Lin, Chih-Jen) [TIST 2011]
> >
> > [4] K-nearest neighbors (Kramer, Oliver and Kramer, Oliver) [Dimensionality reduction with unsupervised nearest neighbors]

---

> > ### Author Response · Authors · 2023-11-21
> > **Response (Part 3/3)**
> >
> > **Q4.3**: The influence of DNN depths to the performance of PTaRL.
> >
> > **R**:
> > Thank you for the insightful suggestion. We employed MLP and FT-Transformer as our two backbone models to examine the impact of DNN depths on the performance of PTaRL.
> > The experiments involved gradually increasing the depths of these backbones, and subsequently evaluating the model performance across three datasets encompassing binary classification, multiclass classification, and regression tasks.
> > The detailed experimental results are presented below and can also be found in Table 11 in the Appendix.
> > $\uparrow$ represents higher evaluation metric is better for classification and $\downarrow$ represents lower evaluation metric is better for regression.
> > The outcomes indicate that, in the absence of PTaRL, augmenting the model depth does not consistently enhance performance; in fact, there is a decline in performance when the depth becomes excessively deep. Contrastingly, when applying PTaRL, the model exhibits remarkable robustness across varying depths. We posit that the utilization of global prototypes for representations in P-Space, as incorporated in the proposed PTaRL, significantly contributes to this observed effect.
> >
> >
> >
> > | Datasets                          | Models                          | layers = 3     | layers = 5     | layers = 7     | layers = 9     |
> > |-----------------------------------|---------------------------------|----------------|----------------|----------------|----------------|
> > | HI $\uparrow$                     | MLP                             | 0.680          | 0.707          | 0.675          | 0.669          |
> > |                                   | MLP + PTaRL                     | **0.719**      | **0.730**      | **0.729**      | **0.732**      |
> > |                                   | FT-Transformer                  | 0.687          | 0.719          | 0.709          | 0.691          |
> > |                                   | FT-Transformer + PTaRL          | **0.738**      | **0.742**      | **0.741**      | **0.737**      |
> > | JA $\uparrow$                     | MLP                             | 0.670          | 0.715          | 0.704          | 0.689          |
> > |                                   | MLP + PTaRL                     | **0.708**      | **0.732**      | **0.728**      | **0.733**      |
> > |                                   | FT-Transformer                  | 0.689          | 0.716          | 0.729          | 0.709          |
> > |                                   | FT-Transformer + PTaRL          | **0.738**      | **0.741**      | **0.745**      | **0.742**      |
> > | CA $\downarrow$                   | MLP                             | 0.522          | 0.513          | 0.509          | 0.524          |
> > |                                   | MLP + PTaRL                     | **0.491**      | **0.480**      | **0.482**      | **0.479**      |
> > |                                   | FT-Transformer                  | 0.486          | 0.476          | 0.472          | 0.478          |
> > |                                   | FT-Transformer + PTaRL          | **0.448**      | **0.446**      | **0.442**      | **0.443**      |
> >
> >
> > We would like to express our sincere gratitude for your insightful suggestions which have significantly contributed to the refinement of our paper. Classical methodologies, such as XGBoost, KNN, and SVM, enjoy widespread adoption among practitioners for solving tabular problems, owing to their commendable performance and user-friendly nature. The comparative analysis between deep learning models and traditional approaches is instrumental in advancing the overall model generality within the tabular domain. We hope that our proposed method could offer valuable insights to the community.

---

> ### Comment · Reviewer_GXco · 2023-11-22
>
> Many thanks to the authors for their thorough and insightful responses. I am happy to increase my score to 8.

---

> > ### Author Response · Authors · 2023-11-22
> > **Response**
> >
> > Thank you for taking the time to review our response and for your positive feedback.

---

### Official Review · Reviewer_z5Xh · 2023-11-01

**Soundness:** 4 excellent
**Presentation:** 3 good
**Contribution:** 4 excellent
**Rating:** 8
**Confidence:** 2

**Summary:**

The existing deep tabular ML models suffer from the representation entanglement and localization. To address this, the authors explore a novel direction of applying prototype learning  framework. The proposed framework involves to construct prototype-based projection space and learn the disentangles representation around global data prototypes.

The proposed method contains two stages: prototype generating and prototype projecting. The former is to constructs global prototypes as the basis vectors of projection space for representation, and the latter is to project the data samples into projection space and keeps the core global data information via optimal transport. The authors show the efficiency of the proposed method with various benchmarks.

**Strengths:**

The proposed approach is novel and the experimental results are impressive.

**Weaknesses:**

It would be great if the authors apply the proposed method to recent deep models for tabular representation, such as SAINT [1].

[1] Saint: Improved neural networks for tabular data via row attention and contrastive pre-training, NeurIPS workshop 2022

**Questions:**

I wonder if the authors believe that the proposed method can be applied to generative models.

---

> ### Author Response · Authors · 2023-11-21
> **Response**
>
> We express our sincere gratitude for your valuable efforts and instructive advice to improve the manuscript. We respond to each of your comments as follows:
>
> **W1**: Further experimental results of applying PTaRL to recent deep models for tabular representation.
>
> **R**: Thanks for your valuable suggestion. We have applied PTaRL to recent deep tabular model SAINT [1], making it one of our baseline models.
> We provide additional experiment results in Section 6 Table 1, Appendix Table 5 and Table 6.
> The main results as follows:
>
>
> |                        | AD $\uparrow$  | HI $\uparrow$  | HE $\uparrow$  | JA $\uparrow$  | AL $\uparrow$  | CA $\downarrow$ | Win |
> |------------------------|----------------|----------------|----------------|----------------|----------------|-----------------|-----|
> | SAINT                  | 0.826          | 0.689          | 0.363          | 0.675          | 0.913          | 0.492           | 0   |
> | SAINT + PTaRL          | **0.861**      | **0.728**      | **0.401**      | **0.728**      | **0.950**      | **0.471**       | **6**|
>
> $\uparrow$ represents higher evaluation metric is better for classification and $\downarrow$ represents lower evaluation metric is better for regression. The results show that PTaRL has the ability to generally improve deep tabular models' performance, including recently proposed models such as SAINT.
>
> [1] SAINT: Improved Neural Networks for Tabular Data via Row Attention and Contrastive Pre-Training (Gowthami Somepalli and Avi Schwarzschild and Micah Goldblum and C. Bayan Bruss and Tom Goldstein) [NeurIPS 2022 Workshop]
>
>
> **Q1**: Can the proposed PTaRL be applied to generative models?
>
> **R**:
> Thanks for your constructive suggestion.
> We hope that our proposed method could offer valuable insights to enhance generative model representations and it will be an important direction for our future work.
> Take the research line of deep generative models for representation learning as an example.
> Deep generative models excel in discerning the true data distribution by learning the distribution of input data within the latent space.
> In this context, the representation of data in the latent space serves as a comprehensive data representation.
> It appears that PTaRL offers a potential solution to incorporate global data information through prototypes.
> Leveraging such global data information to enhance the representation of latent structures aids in better modeling the true data distribution, consequently improving the generative model's performance.

---

### Official Review · Reviewer_7NUi · 2023-11-01

**Soundness:** 3 good
**Presentation:** 2 fair
**Contribution:** 2 fair
**Rating:** 8
**Confidence:** 4

**Summary:**

This paper introduces several techniques to improve the performance of neural networks on tabular data. The study demonstrates that deep tabular models often face issues related to representation entanglement and the loss of global structure. To address these challenges, the paper proposes the construction of a prototype-based projection space with two carefully designed constraints aimed at decoupling the projected representations.

**Strengths:**

- The suggested representation learning pipeline can be integrated into various deep tabular models.
- Figure 1 clearly illustrates that the phenomenon of representation entanglement has not been mitigated as the model capacity is gradually increased.
- The primary concept for enhancing representation involves using weighted prototypes to approximate the original mapped features. This idea is indeed intriguing.

**Weaknesses:**

- The illustration (Fig. 2) does not clearly depict the overall pipeline; it still remains unclear.
- More details of the optimization process could be provided.

**Questions:**

1. How is equation (4) optimized? Compared to the traditional OT problem, it includes \theta_f as a variable to be optimized.
2. Could you provide more technique details about the workflow of the PTARL algorithm (Algorithm 1)?
3. The illustration (Fig. 2) could benefit from improvement as it currently lacks clarity in depicting the overall pipeline. For instance, there are two blocks labeled "Hidden Representation"; could you clarify the distinction between them? Additionally, the three sentences on the right side of the figure require further explanation for better understanding.

---

> ### Author Response · Authors · 2023-11-21
> **Response (Part 1/2)**
>
> We sincerely appreciate your efforts and instructive advice to improve the manuscript. We respond to each of your comments as follows:
>
> **Q1**: Unclear description of the optimization process of Eq. 4.
>
> **R**: We are sorry for the confusion and we would like to give a more clear description.
>
> The Optimal Transport (OT) problem is usually to find the most cost-effective way to transform a given distribution to another distribution, which is typically achieved by calculating the specified transportation plan that minimizes the total transportation cost, while the minimized cost is usually called OT distance.
> In our paper, we minimize the distance between original representation distribution over each sample $P_i$ by deep tabular models and the corresponding projection representation distribution $Q_i$ in P-Space with global prototypes, in order to preserve original data information (of $P_i$) in $Q_i$.
> We follow the typical setting of OT problem to first estimate the transport plan to obtain the OT distance between $P_i$ and $Q_i$.
> Then, the obtained OT distance is further used as loss function to jointly learn the two representations.
>
> Specifically, after initializing the global prototypes $B$ of P-Space, we project the original data samples into P-Space to learn the representations with global data structure information.
> To better illustrate the optimization process, we revise the Eq. 4 in the original paper to make it more readable.
> In Eq. 4, the $i$-th sample representation by deep tabular model is denoted as $G_f(x_i;\theta_f)$, the empirical distribution over this sample representation is $P_i = \delta_{G_f(x_i;\theta_f)}$, the projection representation distribution is denoted as: $Q_i = \sum_{k=1}^{K} r_{i}^{k} \delta_{\beta_{k}}$, where $r_i$ is coordinates.
> To capture the shared global data structure information, we formulate the representation projecting as the process of extracting instance-wise data information by $G_f(x_i;\theta_f)$ to $P_i$, and then pushing $P_i$ towards $Q_i$ to encourage each prototype $\beta_k$ to capture the shared global data structure information, a process achieved by minimizing the OT distance between $P_i$ and $Q_i$.
> The OT distance between $P_i$ and $Q_i$ could first be calculated by: $\textbf{OT}(P_i, Q_i) = \min_{\textbf{T}_i\in \Pi (P_i, Q_i)} \langle \textbf{T}_i, \textbf{C}_i \rangle$,
>
> where $C_{ik} = 1-cos(G_f(X_i;\theta_f), \beta_k)$, the average OT distance between $P_i$ and $Q_i$ over train sets could be viewed as loss function $L_{projecting}(X, B)$ to be further optimized:
>
> $\min\frac{1}{n}\sum_{i=1}^{n} \textbf{OT}(P_i, Q_i) = \min_{\theta_{f}, \gamma, B}  \frac{1}{n}\sum_{i=1}^{n} \min_{\textbf{T}_i\in \Pi (P_i, Q_i)} \langle \textbf{T}_i, \textbf{C}_i \rangle$, we use gradient descent to update $\theta_f, \gamma, B$. We also provide the detailed description in Appendix A.7 in the original paper.
>
> The idea of applying OT distances as loss functions has been employed in multiple applications.
> For example, [1] proposes using this idea for semantic segmentation, and [2] applies this idea to outlier detection in data.
>
> [1] Importance-aware semantic segmentation in self-driving with discrete wasserstein training (Liu, Xiaofeng and Han, Yuzhuo and Bai, Song and Ge, Yi and Wang, Tianxing and Han, Xu and Li, Site and You, Jane and Lu, Jun) [AAAI 2020]
>
> [2] Outlier-robust optimal transport (Mukherjee, Debarghya and Guha, Aritra and Solomon, Justin M and Sun, Yuekai and Yurochkin, Mikhail) [ICML 2021]

---

> > ### Author Response · Authors · 2023-11-21
> > **Response (Part 2/2)**
> >
> > **Q2**: Further explanation on the PTaRL workflow.
> >
> > **R**: We have revised the workflow of PTaRL in Algorithm 1 in the original paper's Appendix to provide more technique details.
> >
> > As shown in the algorithm, Line 1 to 3 represents the global prototype initialization process.
> > Given any deep tabular model $F(\cdot;\theta) = G_h\left(G_f(\cdot;\theta_f); \theta_h\right)$, we first train it by Eq. 2 and then apply K-Means clustering to the output of the trained backbone $G_f(\cdot;\theta_f)$ to get global prototypes $B$.
> >
> > From Line 4 to 19, the second stage of representation projecting is illustrated.
> > The parameters of $F(\cdot;\theta)$ need to be re-initialized.
> >
> > Line 8 to 14 details the process of calculating $L_{projecting}$.
> > Within a minibatch, for each instance, we calculate the OT distance between its representation by the deep tabular model and the projection representation in P-Space.
> > We then average the OT distance of each instance within the minibatch as the $L_{projecting}$.
> > In Line 12, $C_i$ is the transport cost matrix calculated by cosine distance: $C_{ik} = 1-cos(G_f(X_i;\theta_f), \beta_k)$.
> > The $L_{projecting}$ would be further optimized to push $P_i$ towards $Q_i$ to encourage each prototype to capture the shared global data structure information.
> >
> > We introduce $L_{task}$ in Line 15 to incorporate task-specific label supervision.
> > $L_{diversifying}$ in Line 16 is introduced as the P-Space Coordinates Diversifying Constraint to alleviate the collapse of coordinates generating.
> > Additionally, $L_{orthogonalization}$ in Line 17 serves as the Global Prototype Matrix Orthogonalization Constraint, ensuring the independence of global prototypes from each other.
> > Here, $M \in [0, 1]^{K\times K}$ and $M_{ij} = \\|\cos(\beta_i, \beta_j)\\|_1$.
> >
> >
> > **Q3**: Unclear description in the illustration of Fig. 2.
> >
> > **R**: Thanks very much for pointing this confusion. We have revised our Fig. 2 in the original paper.
> >
> > We have clarified the hidden representation in Phase 1 and the projection representation in Phase 2.
> > In phase 1, given any deep tabular model $F(\cdot;\theta) = G_h\left(G_f(\cdot;\theta_f); \theta_h\right)$,
> > we initially train it with Eq. 2. Subsequently, K-Means clustering is applied to the output of the trained backbone $G_f(\cdot;\theta_f)$ to obtain global prototypes $B$ for constructing P-Space.
> > Moving on to Phase 2, $G_f(\cdot;\theta_f)$ and $G_h(\cdot;\theta_h)$ are re-initialized as $G_f(\cdot;\theta_f^*)$ and $G_h(\cdot;\theta_h^*)$.
> > The representation generated by $G_f(\cdot;\theta_f^*)$ is then projected into P-Space to learn the projection representation, denoted as ''Space Calibration'' in Fig. 2.
> > In this process, a shared coordinates estimator calculates coordinates for the projection in P-Space, and then the projection representation is obtained by $\sum_{k=1}^{K} r_{i}^{k} {\beta_{k}}$.
> > The projection representation in P-Space would be finally processed by $G_h(\cdot;\theta_h^*)$ to give predictions.
> >
> > Moreover, we give further explanation about the three sentences on the right side of Fig. 2.
> > ''Projecting Via Optimal Transport'' indicates that the representation by $G_f(\cdot;\theta_f^*)$ is pushed forward to projection representation by minimizing the Optimal Transport Distance between the two representations.
> > The sentences ''coordinates diversifying'' and ''prototypes orthogonalization'' correspond to two constraints for representation disentanglement, i.e. P-Space Coordinates Diversifying Constraint and Global Prototype Matrix Orthogonalization Constraint.

---

> > > ### Comment · Reviewer_7NUi · 2023-11-23
> > >
> > > Thank you for the author's response. The current workflow clearly illustrates the overall process and the computation involved in Equation 4. I would like to increase my rating. Regarding the updated workflow, Step 18 remains unclear. Please make it more understandable.

---

### Meta-Review · Area_Chair_pczb · 2023-12-09

**Metareview:**

This paper addresses the challenges of representation entanglement and localization in existing deep tabular machine learning methods. To overcome these issues and enhance prediction performance, the authors propose PTaRL, a prototype-based tabular representation learning framework. PTaRL constructs a prototype-based projection space, involving prototype generation and prototype projecting stages, and introduces constraints for representation diversification and prototype entanglement avoidance. Experimental results demonstrate the consistent superiority of PTaRL when coupled with state-of-the-art deep tabular machine learning models across various tabular benchmarks.

Three reviewers have evaluated this paper, unanimously agreeing that the paper's ideas are clear and convincing, and the experimental results are impressive. They strongly advocate for unanimous acceptance. The authors handle a crucial issue where deep learning models for tabular data fail to learn disentangled representations, unlike in domains such as images, which I think leads to the failure of of many deep learning techniques in diverse tasks for tabular data. I think the proposed method, PTaRL, shows significant effectiveness across various architectures, including the application to state-of-the-art deep learning models, suggesting that this method can provide valuable insights for applying well-known techniques from other domains in diverse learning settings for tabular data beyond the scope of this paper. While the paper is generally smooth to read, there is a slight ambiguity above eq (4), particularly regarding the architecture/implementation of phi, the definition of 'global data structure information,' and the justification for using shared phi to project representations and applying OT etc. Addressing these concerns could further enhance the paper's overall quality.

**Justification For Why Not Higher Score:**

n/a

**Justification For Why Not Lower Score:**

The paper introduces a method that can enhance the representation of deep models handling tabular data, anticipating a significant impact and insight for various tabular tasks in the future.

---

### Decision · Program_Chairs · 2024-01-16

Accept (spotlight)